# Lung Metastatectomy: Can Laser-Assisted Surgery Make a Difference?

**Konstantinos Grapatsas** [1], **Anastasia Papaporfyriou** [2] , **Vasileios Leivaditis** [3], **Benjamin Ehle** [4]
**and Michail Galanis** [5,*]

1   Department of Thoracic Surgery, University Clinic Ostwestfalen-Lippe, Campus Bielefeld,
    33604 Bielefeld, Germany
2   Division of Pulmonology, Department of Internal Medicine II, Medical University of Vienna,
    1090 Vienna, Austria
3   Department of Cardiothoracic and Vascular Surgery, Westpfalz-Klinikum, 67655 Kaiserslautern, Germany
4   Department of General, Visceral, Thoracic and Oncological Surgery, Helios Amper Hospital Dachau,
    85221 Dachau, Germany
5   Department of Thoracic Surgery, Inselspital, Bern University Hospital, University of Bern,
    3012 Bern, Switzerland
*   Correspondence: michail.galanis@insel.ch; Tel.: +41-31-632-37-45

**Abstract:** Background: Resection of lung metastases with curative intention in selected patients is associated with prolonged survival. Laser–assisted resection of lung metastases results in complete resection of a high number of lung metastases, while preserving lung parenchyma. However, data concerning laser lung resections are scarce and contradictory. The aim of this study was to conduct a systematic review to evaluate the utility of laser-assisted pulmonary metastasectomy. Methods: An electronic search in MEDLINE (via PubMed), complemented by manual searches in article references, was conducted to identify eligible studies. Results: Fourteen studies with a total of 1196 patients were included in this metanalysis. Laser-assisted surgery (LAS) for lung metastases is a safe procedure with a postoperative morbidity up to 24.2% and almost zero mortality. LAS resulted in the resection of a high number of lung metastases with reduction of the lung parenchyma loss in comparison with conventional resection methods. Survival was similar between LAS and conventional resections. Conclusion: LAS allows radical lung-parenchyma saving resection of a high number of lung metastases with similar survival to conventional methods.

**Keywords:** morbidity; pulmonary metastasectomy; laser-assisted; laser; lung metastases; metastases; postoperative complications





## 1. Introduction

Approximately 30% of patients with malignant tumors will develop pulmonary metastases. Pulmonary metastasectomy (PM) is an established treatment with low morbidity and mortality rates in selected patients with lung metastases [1–3]. These patients may benefit from a lung resection with curative intent as part of an oligometastatic concept. A PM is generally indicated if the primary tumor can be well controlled and the patient has sufficient cardiopulmonary reserves [4–6]. In 1997, the results of the International Registry of Lung Metastases from 5206 patients from Canada, the United States and Europe showed evidence of survival prolongation after PM [7]. However, no prospective randomized trials have compared PM with other therapeutic options. Nevertheless, surgical resection of pulmonary metastases is widely performed as part of the treatment for various primary tumors. The 5-year survival rate after PM ranges from 20% to 80% [4–6].

Pulmonary metastasectomy is nowadays generally accepted as a standard treatment if the following conditions are established: (a) the primary malignancy is controlled or concontrollable; (b) if there is another evidence of metastatic disease, this should be controllable; (c) the patient must have a low risk for surgical intervention, and (d) a complete

resection of all pulmonary metastases should be possible. In addition, PM may be indicated for establishment of histology/re-histology in cases where the primary tumor is well controlled but there is the possibility of neoplastic transformation and thus a different histological entity [8]. In the last few years, indications for lung metastasectomy have not changed. However, technological evolution has allowed the development of new surgical instruments and techniques. As a result, the resection of lung metastases can be performed with various surgical instruments such as staplers, electrocautery devices, ligasure/ultracision devices and Nd:YAG lasers. Laser-assisted surgery (LAS) was introduced in the last twenty years for the resection of lung metastases [5–7]. The first large series with patients undergoing LAS for the treatment of pulmonary metastases were described by Mineo et al. and Rolle et al. in 1994 and 2006, respectively [9,10]. In the following years, several studies described the surgical benefits of LAS in PM. Pulmonary LAS allows the resection of large numbers of lung metastases with minimal morbidity and mortality rates. In addition, in the case of pulmonary recurrence, a repeated metastasectomy can be performed [10–15]. Unfortunately, all the existing studies concerning pulmonary LAS for lung metastases are single-centered and describe the experiences. The aim of the current review and meta-analysis is to investigate cumulative possible indications and benefits for LAS in lung metastases.

## 2. Materials and Methods

This systematic review and meta-analysis was performed according to the Preferred Reporting Items for Systematic Reviews and Meta-Analyses (PRISMA) Statement protocol [16]. Quality assessments of individual studies were conducted using the Newcastle-Ottawa Scale [17].

### 2.1. Search Strategy and Organisation

We searched MEDLINE (via PubMed) from 1985 up to 2020 to identify relevant studies with the theme under review. Our search included the following subject headings and/or key words variably combined: "Nd:YAG laser", "laser", "pulmonary resection", "lung resection", "lung metastasis", and "pulmonary metastasis". Reference lists of articles initially extracted from bibliography were searched by hand to identify additional relevant reports. The eligibility of references retrieved by this search and the risk of bias for each retrieved study were assessed independently by two authors (Dr. Grapatsas and Dr. Leivaditis). The two authors collected data from each repost independently. The authors resolved differences of opinion by appeal to a third review author (Dr. Ehle) when necessary. Full texts of the remaining retrieved articles were examined independently from the two authors (K.G. and V.L) to determine whether the articles contained relevant information. The inclusion of the final retrieved articles was approved by all authors. During the literature research, special concern was paid to identifying the number of resected lung metastases and reported postoperative complications.

### 2.2. Inclusion and Exclusion Criteria

Studies were considered eligible if they included human patients undergoing LAS for lung metastases with curative intent.

### 2.3. Studies Were Excluded Based on Any of the Following Criteria

(I)    Reviews, letters, laboratory research, animal experiments were excluded;
(II)   The language of the study was not English;
(III)  Studies that examined resection techniques other than LAS for lung metastases.

### 2.4. Quality Assessment

Each included study's quality was assessed using the Newcastle–Ottawa Scale (NOS). Based on the quality of selection, comparability and exposure, a score with a maximum of nine points was appointed.

### 3. Results

After primary retrieval in Medline, a total of 556 potentially relevant studies were incorporated into our initial search. Of these, 502 of articles were excluded as irrelevant from the title or abstract screening. Full texts were retrieved from the remaining 54 studies. Fourteen of them met all the inclusion criteria in the analysis (Table 1).

#### 3.1. Characteristics and Qualities of the Included Studies

Studies that were included are summarized in Table 1. A total of 1196 patients participated in these studies. There were five studies with less than 50 patients [14,18–22]. Twelve studies were retrospective cohort studies. One study was prospectively designed [23]. Only one study was a randomized perspective trial [19]. Six of them were published after 2017. Six studies compared LAS for pulmonary metastases with conventional method resections. Nine studies took place in Germany and three in Italy (Table 1). Quality assessments of individual studies are shown in Table 1.

#### 3.2. Study Population Characteristics

Ten studies investigated each center's experience with LAS from various primary tumors. One study investigated the joint experience of two centers [23]. Only three studies investigated the results after pulmonary LAS from specific tumors. One study investigated the results of LAS from renal cell carcinoma [24], one from colorectal cancer [13] and one from osteosarcoma and soft-tissue-sarcoma [14]. The age of patients ranged from 13 to 86 years.

#### 3.3. Surgical Characteristics

LAS was performed in 1174 cases with muscle sparring anterolateral thoracotomy. In 22 cases, LAS was performed using video-assisted thoracic surgery (VATS), and in 99% ($n = 1163$) of cases there were laser-assisted wedge resections or segmentectomies. Lobectomies for the resection of the lung metastases were required only in a small percentage of patients (around 1% ($n = 11$)) [13,25]. In some studies, the need for lobectomy was reduced up to 95%, even when metastases were centrally located [25].

#### 3.4. Complication Rates and Mortality

The postoperative complication rates were between 1.2% and 24.2% [10,12,26]. Postoperative complications described were pneumothoraxes, hemothorax, lung atelectasis, persistent air leak and pneumonia. The most frequent complication was postoperative pneumonia, which was diagnosed in up to 11.3% of patients. Persistent air leak was reported in up to 8.4%. Cases of postoperative hemothorax were limited [10]. Postoperative mortality was very low, and reported in 11 studies to be 0% (Table 1).

#### 3.5. Rates of Complete Resection, Local Relapse and Survival with LAS

A R0-resection with LAS was achieved in 62.9% to 100% of lung metastases resection [10,21]. Local relapse was reported up to 0.8% [27].

#### 3.6. Survival

The laser-assisted resection of lung metastases from various primary tumors resulted in a 5-year-survival from 35% to 65.7%. Incomplete resection of the metastases with the laser was associated with poor prognosis [24].

**Table 1.** Basic characteristics of the included studies.

| Study | Patients Source | Study Period | Laser-Pattern | Follow-Up Duration (Range), Months | Mean Age (Range) Years | Patients with LAS | Histological Type | R0-Resection-rate | Mortality | Morbidity | 5-Year-Survival | Notes | NOS |
|---|---|---|---|---|---|---|---|---|---|---|---|---|---|
| LoCicero III et al, 1989 [18] | USA | 1985–1988 | 1064-nm Nd:YAG | 6 | - | 10 | VPT | - | - | - | - | | 6 |
| Mineo, 1998 [19] | Italy | 1987–1995 | Nd:YAG | - | 56.8 (13–77) | 23 | VPT (24.4% CRC) | - | - | - | - | - LAS vs. diathermic device<br>- randomized perspective trial<br>- LAS allowed more tissue sparing<br>- LAS reduced:<br>- hospital stay<br>- postoperative air leakage | 8 |
| Rolle, 2002 [25] | Germany | 1996–1998 | 1318-nm Nd:YAG | 26.5 | 60 | 100 | VPT (25% CRC, 29% RCC) | 97.5% | - | - | 32% | - 41% of metastases located centrally-> in 95% LAS resection possible, -only in 5% lobectomy necessary | 7 |

**Table 1.** *Cont.*

| Study | Patients Source | Study Period | Laser-Pattern | Follow-Up Duration (Range), Months | Mean Age (Range) Years | Patients with LAS | Histological Type | R0-Resection-rate | Mortality | Morbidity | 5-Year-Survival | Notes | NOS |
|---|---|---|---|---|---|---|---|---|---|---|---|---|---|
| Rolle, 2006 [10] | Germany | 1996–2003 | 1318-nm Nd:YAG | 31 (1–198) | 61 (20–80) | 328 | VPT (34% RCC, 28% CRC) | 84.8% | 0% | 1.2% | 35% | - incomplete resection -> reduced survival | 7 |
| Osei-Agyemang, 2013 [12] | Germany | 2005–2010 | 1318-nm Nd:YAG | 27.2 (2.3–60.6) | 64 (11–86) | 62 | VPT | 62.9% | 0% | 24.2% | - | - suspicion of more R1-resections with LAS<br>- no lobectomy or pneumonectomy needed with LAS<br>- more often pneumonia after LAS | 7 |
| Baier, 2015 [24] | Germany | 1996–2012 | 1318-nm Nd:YAG | 46 (2–198) | 63 | 237 | RCC | 95.3% | 0% | - | 54% | - all complications -><br>- conservatively complete resection of metastases -> besser OS (54% vs. 7%) | 7 |

**Table 1.** *Cont.*

| Study | Patients Source | Study Period | Laser-Pattern | Follow-Up Duration (Range), Months | Mean Age (Range) Years | Patients with LAS | Histological Type | R0-Resection-rate | Mortality | Morbidity | 5-Year-Survival | Notes | NOS |
|---|---|---|---|---|---|---|---|---|---|---|---|---|---|
| Franzke, 2016 [27] | Germany | 2010–2015 | 1318-nm Nd:YAG | 23.8 (2–66) | 59.3 (17–85.4) | 99 | VPT | 0% | 0% | 13% | 65.7% | - More metastases resected with LAS<br>- Trend to local relapse after NLAS (0.8% vs 3.1%) | 7 |
| Schmid, 2017 [14] | Germany | 2006–2016 | 1320-nm Nd:YAG | - | 43.6 | 29 | osteosarcoma, soft-tissue-sarcoma | 72% | 0% | 22% | 53% | - more metastases resected with LAS<br>- more minor complications with LAS<br>- OS similar | 7 |
| Porello, 2018 [28] | Italy | 1995–2009 | 1318-nm Nd:YAG | 72 (3–72) | 62 | 106 | VPT | - | 0% | - | 46% | - 8.7% prolonged air leak<br>- Re-LAS by recurrence | 7 |

**Table 1.** *Cont.*

| Study | Patients Source | Study Period | Laser-Pattern | Follow-Up Duration (Range), Months | Mean Age (Range) Years | Patients with LAS | Histological Type | R0-Resection-rate | Mortality | Morbidity | 5-Year-Survival | Notes | NOS |
|---|---|---|---|---|---|---|---|---|---|---|---|---|---|
| Moneke, 2019 [13] | Germany | 2005–2016 | 1320-nm Nd:YAG | 53 | 63 (33–81) | 77 | CRC | 92% | 0% | 22% | 51% | - LAS:<br>- more metastases resected<br>- fewer anatomical resections needed | 6 |
| Stefani, 2019 [20] | Italy | 2005–2017 | 1318-nm Nd:YAG | 49 (7–79) | 65 (42–79) | 42 | VPT | 91% | 0% | 29% | 66% | - LAS for large or central metastases<br>- Comparison LAS vs lobectomy<br>- with lobectomy longer operative time, chest drain and hospital stay, more complications (29% vs. 40%)<br>- n.s. in OS | 7 |
| Meyer, 2018 [21] | Germany | 2014–2018 | 1320-nm Nd:YAG | 6 | 60 | 15 | VPT | 100% | 0% | 0% | - | VATS-LAS | 6 |

| Study | Patients Source | Study Period | Laser-Pattern | Follow-Up Duration (Range), Months | Mean Age (Range) Years | Patients with LAS | Histological Type | R0-Resection-rate | Mortality | Morbidity | 5-Year-Survival | Notes | NOS |
|---|---|---|---|---|---|---|---|---|---|---|---|---|---|
| Mc Loughlin, 2018 [22] | Ireland | 2017 | 1318-nm Nd:YAG | - | 61 (46–67) | 7 | VPT | 100% | 0% | - | - | - VATS-LAS<br>- 1 patient with prolonged air leak | 6 |
| Hassan, 2021 [23] | Germany | 2017–2019 | 1320-nm Nd:YAG | 6 | 64 (20–88) | 61 | VPT | - | 0% | 8.1% | - | - recovery of the lung function is associated with the number of resected metastases and their depth | 8 |

*3.7. Comparison between LAS and Conventional Resection Techniques*

Compared to conventional methods, LAS resulted in the resection of a higher number of lung metastases (Osei-Agyemang et al.: mean seven lung metastases resected with LAS vs. two with conventional methods; Frantzke et al.: more than two resected metastases: 30.3% with LAS vs. 6.3% with conventional methods) [12,27]. In almost all studies both techniques achieved similar R0 resections. [13–27]. Additionally, in the conventional group, a greater number of lobectomies for complete resection of lung metastases were needed [12,13]. LAS also allowed more tissue sparring resections when compared with diathermic devices (mean ratio lesion diameter/volume of resected lung metastasis: 0.94 vs. 1.11 with conventional methods) [9,19,29]. Furthermore, a trend to local relapse after LAS was lower compared with conventional resection methods (0.8% vs. 3.1%) [27]. However, the rate of postoperative complications varied among studies. For example, in the study of Schmid et al., more minor complications were detected after LAS [14]. On the other hand, Stefani et al. reported longer operative times for anatomical resections (143 min vs. 114 min with LAS), a longer requirement for chest drain (3.3 days vs. 2.2 days with LAS) and longer postoperative stays (6.0 days vs. 4.3 days with LAS) [20]. Osei-Agyemang reported no difference concerning complications after conventional resections and LAS-resection. In this study, no difference was found concerning persistent air leak (1.3% vs. 0% with LAS), need for surgical revision (1.7% vs. 3.2%), placement of new chest drain postoperative because of pneumothorax or pleural effusion (2.9% vs. 1.6%) and postoperative pleural empyema (1.3% vs. 1.6%). On the contrary, in the same study after LAS, an increased risk for postoperative pneumonia was detected (2.9% vs. 11.3% for LAS). 5-year-survival for LAS ranged from 53% to 70% and for conventional resections from 38% to 73.6% [10,12–14,20].

## 4. Discussion

LAS is a safe method with almost zero mortality and minor morbidity. LAS allows the complete erasure of deep-seated lesions, while sparing the lung parenchyma, reducing the need for lobectomy and resecting only the needed tissue. Most of the identified studies in the literature describe a single-center experience with laser-assisted PM. Cohort studies that compare LAS with conventional resections are limited [12,13,27]. Pre-existing associated reviews and letters to editor are also rare. Panagiotopoulos et al. and Ojanguren et al. described experiences with LAS and VATS-LAS in their editorial and review articles [30,31]. Marscherey et al. described in their review article in the German language experiences up to 2016 with LAS and PM [32]. Due to the nature of these studies, they were excluded from the current metanalysis.

Developing a laser device for lung resections faced initial hesitancy and failures. Lung tissue has unique features, as it contains water in a high percentage of around 80% and has a high vessel density, but low tissue density. To avoid bleeding, or a persisting air leak, a device with precise cutting and coagulation capabilities is needed. Historically, the first experience with laser-assisted pulmonary metastasectomy was described in 1967 by Minton et al. [11]. The discussion in using a laser device for resection of PM was again reopened in 1985 by LoCicero et al. The authors used a CO2 laser that was considered inadequate for PM [18]. However, after the development of multiple laser devices, centers in Europe, Japan and USA began performing laser-assisted PM with 1064 Nd:YAG. Finally, in 2007, the development of the 1318 Nd:YAG, that could deliver 100 Watts, allowed resections of many lung metastases while reducing postoperative thermal tissue edema [33].

Today, pulmonary laser-assisted metastasectomy is a well-established method. In laser metastasectomy, an anterolateral muscle-sparing thoracotomy is usually performed [14,15]. Bilateral LAS can enable the resection of a larger total number of metastases [15]. Resection of bilateral lung metastases by sequential thoracotomy is suggested with a time window of three weeks. All visible and palpable nodules are recognized and resected with tumor margins of 2–3 mm and an additional 5 mm of residual lung parenchyma necrosis. In comparison to other conventional resection instruments, such as electrocautery, ultrasonic or ligasure devices, the Nd:YAG laser allows the resection of metastases from the periphery

to center of the lobe without bleeding [33]. Exposed segmental bronchial branches and blood vessels are sutured. The lung architecture is reconstructed and the visceral pleura is re-approximated with an absorbable suture [32,33]. (Figures 1 and 2).

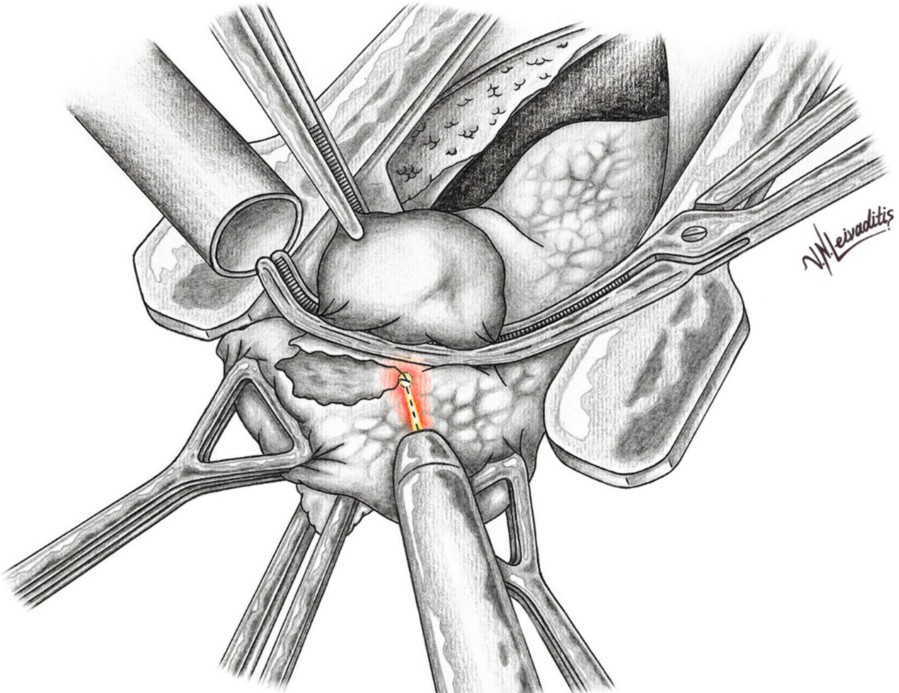

**Figure 1.** The lung metastasis has been isolated from the lung parenchyma with an isolation-clamp. The lung metastasis is resected with the laser-device. The smoke from the laser-resection is vacuumed through the large-lumen smoke-extractor.

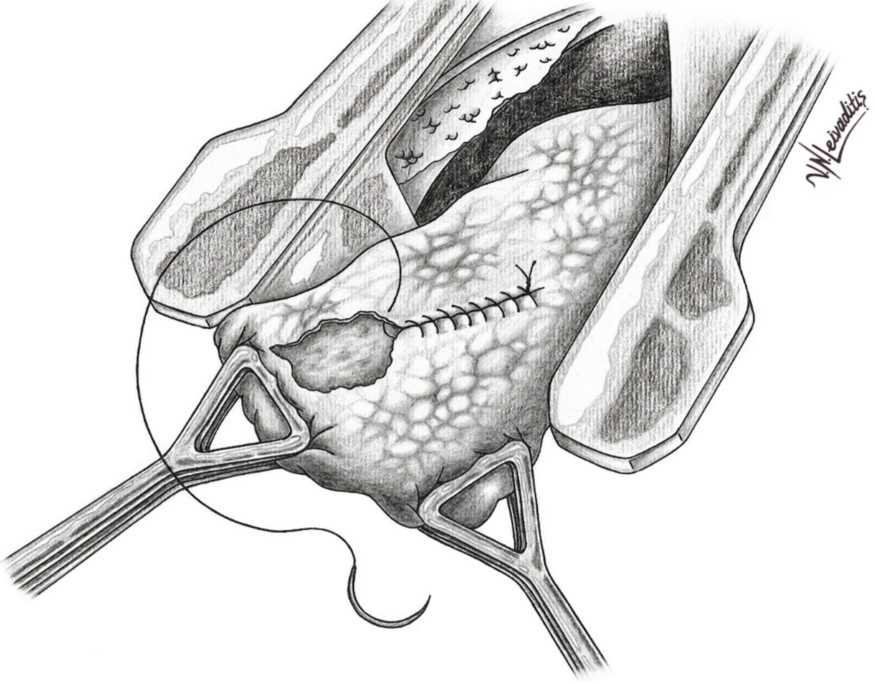

**Figure 2.** The lung parenchyma is reconstructed after the resection of the lung metastasis resection by re-approximating the visceral pleura with an absorbable suture.

In recent years, LAS has been adopted in VATS. VATS-LAS for the resection of lung metastases has been described by Meyer et al. and Mc Loughlin et al. in a limited number of patients. Though results could be questioned due to limited number of cases, the authors showed adequate resections with low complication rates and short postoperative hospitalization. These positive results could be connected with VATS and mini-thoracotomy (5–7 cm) as a less invasive procedure that allowed the palpation of the whole lung resulting in no local recurrence after a short follow-up period (VATS LAS) [21,22]. Regarding local tumor relapse after PM with video-assisted thoracic surgery (VATS), Abdolnour-Bechtold et al. reported similar findings. In this study, only one patient presented with relapse in the stapler line [34]. However, the number of lung metastases that could be resected with VATS was lower, and they were usually located more peripherally, where palpation of the lung for the identification of additional pulmonary nodules was not always possible [27,34].

Pulmonary LAS, according to evidence, is a safe technique. Complications are mainly minor. Schmid et al. reported a trend to minor complications with LAS in comparison to conventional methods (22% vs. 5%). On the other hand, Stefani et al. reported a lower rate for complications in an LAS group than in a conventional group (29% vs. 40%, respectively). Osei-Agyemang et al. reported more cases of postoperative pneumonia after LAS (11.3% vs. 29% with conventional methods), without, however, distinguishing between a genuine bacterial pneumonia or thermal induced pneumonitis from the laser [12]. Moreover, in the same study, no significant difference was shown between LAS and conventional methods concerning prolonged air leak, need for a revision-operation or postoperative pleural empyema. In addition, LAS was related with very low mortality. More specifically, in 11 studies, the mortality rate after LAS was zero (Table 1).

Knowing the negative impact of a lobectomy on patients' health-status [20], LAS has been established as a lung sparing surgical technique and has contributed in the eduction of lobectomies performed for the complete resection of lung metastases. According to Schmid et al. and Osei-Agyemang et al., no lobectomies were necessary in the LAS group, while Franzke et al. showed that LAS reduced the need for a lobectomy by 10%. This percentage was even higher in the trial of Rolle et al. from 2002, in which, despite the fact that 41% of lung metastases were centrally located, a lobectomy was unavoidable only in 5% of cases. Regarding the postoperative course of patients with centrally resected lung metastases with LAS, Stefani et al. reported that not only did LAS reduce the number of lobectomies for large or central metastases, but there were even fewer complications and shorter hospital stays. In line with these reports, LAS could be a future tool for resecting lung metastases in patients with poor residual lung function that would need a lobectomy but at the present time are considered inoperable [33].

Although, in many cases, patients in the LAS group had already negative predictive factors, such as a high number of lung metastases that often were bilateral, a similar OS was reported for both LAS and conventional surgical techniques. As a result, it can be assumed that LAS allows the resection of multiple lung metastases not possible with other techniques. Moreover, it can be hypothesized that LAS could prolong survival through the complete resection of multiple metastases. However, it should be mentioned that these assumptions/conclusions are made from retrospective studies that describe only single center experiences with a high possibility of selection bias. In addition, the study populations included patients with various primary tumors that could affect OS. On the other hand, as a consequence of preserving lung parenchyma even in the cases of resection of multiple metastases, LAS can allow repeated pulmonary metastasectomy even on the same operated lung. In this case, the tumor disease is again controlled with the complete resection of lung metastases [3].

The purpose of PM is the complete resection of lung metastases, which is associated with prolonged survival [3,7]. The clinical significance of minimal pulmonary residuals after PM is not known [27]. Osei-Agyemang et al. reported that the 'no-contact' pulmonary LAS leaves 5 mm thick carbonized tissue at the pulmonary parenchyma as a safety margin. This carbonized tissue can be delineated in three distinct zones: the vaporization zone,

coagulation zone (necrosis) and hyperemia zone [12]. We believe that in this 5 mm safety margin, possible minimal pulmonary metastatic residuals are destroyed. In addition, the low number of local relapse events could be attributed to this safety margin. Furthermore, to minimize local relapse and achieve a R0 resection, Rolle et al. suggested a 3 mm safety margin and an additional 5 mm coagulation zone [14]. These suggestions are further supported by the results of Franke et al., in which the rate of local lung recurrences was lower after LAS than in the conventional metastasectomy group (0.8% vs. 3.1%) [27]. Moreover, Rolle et al. suggested that in the case of doubt, small satellite metastases can be obliterated with a laser [15]. Despite the local carbonization of the lung parenchyma with a laser, it was shown that no thermal damage to the surrounding lung tissue was produced. Kirschbaum et al. reported that monopolar cutters caused more adjacent tissue injuries than lasers [35]. In addition, lung resections carried out using lasers offer airtightness without damaging airways and blood vessels [36,37]. We believe that due to the aforementioned advantages of pulmonary LAS, the R0 resection of a high number of metastases with a low local relapse rate can be possible. In addition, the need for anatomical resection to achieve these goals is limited. Regarding the effect on lung function after LAS, Hassan et al. showed that resection of two or fewer metastases presented a recovery of lung function after 3 months regarding DLCO. However, a decline of DLCO in the whole cohort correlated with the number of resected metastases at 3 months and at 6 months, as well as the depth of metastases in parenchyma. These results are comparable with the findings for the other lung metastases resection techniques [23].

### 4.1. Limitations

The most important limitation of this meta-analysis is the heterogeneity of study-populations. Apart from three studies [13,14,24] that focus on the resection of lung metastases with LAS for specific primary tumors, the rest of the studies examined the results of LAS for various primary tumors. As a result, a clear conclusion on the effect of the technique on OS cannot be made. Moreover, due to the retrospective nature of almost all studies, and the fact that they refer to only one center's experience, bias exists in all of them. As additional bias, in some centers LAS is established as the first option for LM. For this reason, any further statistical analysis of our results was waived, as it would result in possible false conclusions [14]. Moreover, as all studies were retrospective single-center studies, identification of the extent of statistical heterogeneity of the included studies or further exploration of possible causes of heterogeneity among the studies was not possible. Furthermore, in the studies in which LAS was compared with conventional methods, no distinction between the resection devices was made. For example, in the last 15 years, lung metastasectomies with LigaSure Vessel Sealing-System and Ultracision scalpel have been described rarely in the literature. The devices have shown excellent outcomes in resecting lung metastases, although the resection of deep (in lung parenchyma) located metastases is questionable [38–42]. Therefore, we believe that the results of our meta-analysis should be careful interpreted. The number of 1196 patients undergoing LAS represents a characteristic sample of everyday clinical praxis. Thus, we suggest that LAS is an excellent option for selected patients with multiple lung metastases, that could prolong their survival, allowing repeated resection of a metastatic disease. Nevertheless, each decision and selection of resection technique should be made according to the experience of each surgeon and the location and number of lung metastases.

### 4.2. Future Directions

The Nd-YAG laser has a significant influence on sparing lung parenchyma for PM. In addition, the adaption of LAS in VATS could result in resecting a high number of lung metastases with minimal invasion, which in the past would require a thoracotomy. This could attribute to reduction of hospitalization, as well as postoperative pain. Furthermore, despite the initial financial investment to acquire the laser device, LAS could reduce the

costs of the health care system. The only limitation is probably staff education and training in security of this technique [28,33,43].

## 5. Conclusions

Laser-assisted pulmonary metastasectomy is a safe technique that appears to facilitate the resection of a significantly higher number of lung metastases with acceptable minor complications, while resulting in prolonged survival. As a plus, LAS allows the preservation of more lung parenchyma, that can be of use in the case of repeated resections due to recurrences, even on the same lung.

**Author Contributions:** Conceptualization, K.G. and B.E.; methodology, K.G.; software, K.G.; validation, K.G., B.E. and V.L.; formal analysis, K.G.; investigation, K.G. and V.L.; data curation, B.E.; writing—original draft preparation, K.G. and A.P.; writing—review and editing, K.G., B.E. and A.P.; visualization, V.L.; supervision, M.G.; project administration, K.G. and A.P. All authors have read and agreed to the published version of the manuscript.

**Funding:** This research received no external funding.

**Data Availability Statement:** The data presented in this study are available upon request from the corresponding author.

**Conflicts of Interest:** The authors declare no conflict of interest.

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
