# Peer review of "Lung Metastatectomy: Can Laser-Assisted Surgery Make a Difference?"

_curroncol, doi:10.3390/curroncol29100548_

Round 1

Reviewer 1 Report

Dear Editor and Authors, 

It was my pleasure to review this work titled “Lung metastatectomy: can Laser-assisted surgery make the difference?” by Dr. Grapatsa and collegues from Germany, as it is a subject I am quite interested in. 

In this proposed meta-analysis the authors pool and analyze the results of 14 studies of laser assisted lung metastasectomy published in the literature. This study has an interesting premise and I was quite disappointed to see it did not reach its full potential. Specifically, the authors although they have gone through most of the steps of a meta-analysis have not in actuality performed one!! This is disappointing considering the number of patients they have been able to collect (1196) and how beneficial for the thoracic surgical community a true meta-analysis with full statistical analysis would be!! I agree with the authors that most reports are from single institutions and case series, therefore a meta-analysis on the subject is lacking in the literature!! 

As such, I would recommend to the authors (since they have performed the majority of the “hard work” to do the analysis and re-present their manuscript for review!! Please the also my comments below. 

Comments: 

1. I am not sure if the claim made by the authors “It is believed that 15-50% of thoracic surgical resections in Europe consist of PM.” Is accurate. There is no reference given and from personal experience the range can vary but I am certain it does not reach 50% of the caseload of a thoracic surgery department. The authors should re-evaluate this claim and accurately support it! 

      2. One other indication for a metastasectomy the authors do not list is for establishment of histology/re-histology in cases where the primary tumour was well controlled but there is the possibility of neoplastic transformation and thus a different histology may be in play. For example when metastasis occurs some time since initial threatment! 

3. Should the patients which underwent lobectomy (n=11) not be excluded from the analysis as they are not truly small resections? 

4. Should the 5 studies with less than 50 patients not be excluded from the analysis? This is the norm in meta analysis methodology! 

5. How can the authors claim that “In all studies OS (overall survival) was similar between LAS and conventional resections.” since no statistics were performed! 

6. The discussion is thorough and well presented. I particularly enjoyed the supplied pictures which are informative and well drawn. 

7. The manuscript is well written in clear and understandable language with only minor corrections needed. 

Author Response

s. file

Reviewer 2 Report

I would like to congratulate the authors for their interesting and informative paper.

This is a systematic review of studies investigating laser-assisted pulmonary metastasectomy. After searching MEDLINE, the authors identified 14 relevant studies that included a total of 1196 patients who underwent laser-assisted resection of lung metastases. Laser-assisted pulmonary metastasectomy resulted in the resection of a large number of metastases with preservation of lung parenchyma, while conferring similar survival rates compared to conventional resection methods. Therefore, the authors conclude that laser-assisted pulmonary metastasectomy is a safe and effective procedure.

Here, I have made some suggestions that (in my opinion) could help to improve the overall quality of the manuscript.

·         The authors may consider identifying the report as a systematic review in the title of the manuscript.

·         The authors claim that a meta-analysis has been performed. However, a quantitative synthesis is not evident from the reported results. They may wish clarifying this.

·         The authors may consider specifying the methods used to collect data from reports, including how many reviewers collected data from each report and whether they worked independently.

·         The authors may consider listing and defining all outcomes and other variables for which data were sought. They may also consider describing any assumptions made about any missing or unclear information.

·         The authors may consider specifying how many reviewers assessed each study for risk of bias and whether they worked independently.

·         The authors may consider specifying for each outcome the effect measures used in the presentation and synthesis of results.

·         The authors may consider providing more information regarding the synthesis methodology (e.g., processes to decide which studies were eligible for each synthesis, handling of missing summary statistics, methods to identify the presence and extent of statistical heterogeneity, methods to explore possible causes of heterogeneity among study results, sensitivity analyses to assess robustness of the synthesised results).

·         Table 1 has not been provided, and thus it is difficult to assess the quality of presentation of the results.

·         The authors may consider avoiding definitive statements regarding the superiority of laser-assisted pulmonary metastasectomy and provide a more balanced interpretation of the results considering the low level of evidence of the included studies.

·         The authors may consider providing more information regarding the observed complications after laser-assisted pulmonary metastasectomy.

·         The authors may consider discussing any limitations of the review processes used.

·         The authors state that laser-assisted pulmonary metastasectomy is a minimally invasive technique; however, it is mostly performed through thoracotomy. They may consider clarifying this.

Author Response

s. file

Round 2

Reviewer 2 Report

Thank you for taking the time to consider my suggestions and revise your manuscript accordingly.